# Molecule Generation by Heterophilous Triple Flows

## Abstract

Generating molecules with desirable properties is key to domains like material design and drug discovery. The predominant approach is to encode molecular graphs using graph neural networks or their continuous-depth analogues. However, these methods often implicitly assume strong homophily (*i.e.*, affinity) between neighbours, overlooking repulsions between dissimilar atoms and making them vulnerable to oversmoothing. To address this, we introduce *HTFlows*. It uses multiple interactive flows to capture heterophily patterns in the molecular space and harnesses these (dis-)similarities in generation, consistently showing good performance on chemoinformatics benchmarks.

## 1 Introduction

Identifying molecular candidates with specific chemical properties is an integral task in important biochemistry domains such as material design and drug discovery. However, traditional methods rely on expensive exploratory experiments that involve time and resource-intensive investigations (Paul et al., 2010), hindered by the inherent discreteness of the search space and its vast combinatorial possibilities (Reymond et al., 2012; Polishchuk et al., 2013). Deep generative models can employ effective inductive biases to encode molecules and expedite the discovery process by narrowing down the search space; *e.g.*, they have recently shown significant potential for suggesting promising drug candidates *in silico* (Ingraham et al., 2019; Polykovskiy et al., 2020).

Molecules can be presented as input to a deep learning model in different formats. Initial works, *e.g.*, Kusner et al. (2017), Dai et al. (2018) posed molecular generation as an autoregressive problem, utilizing SMILES (short for 'Simplified Molecular-Input Line-Entry System'), *i.e.*, a unique sequence representation for molecules (Landrum et al., 2013). However, the mapping from molecules to SMILES is not continuous, so similar molecules can be assigned vastly different string representations. Graphs provide an elegant abstraction to encode the interactions between the atoms in a molecule, so powerful encoders based on graph neural networks (GNNs, Scarselli et al., 2009; Kipf & Welling, 2017; Veličković et al., 2018; Xu et al., 2019; Garg et al., 2020) have been adopted in recent years. A range of deep learning frameworks have been integrated with GNNs for molecule generation, including, adversarial models (De Cao & Kipf, 2018; You et al., 2018), diffusion models (Hoogeboom et al., 2022), energy-based models (Liu et al., 2021b), and Neural ODEs (Verma et al., 2022) and other flow-based models (Shi et al., 2019; Luo et al., 2021; Zang & Wang, 2020).

We seek to illuminate, and address, a key issue that has been overlooked while using GNNs in molecule generation settings. Standard GNNs employ local message-passing steps on each input graph to exchange information between nodes and their neighbours; implicitly assuming strong *homophily*, *i.e.*, tendency of nodes to connect with others that have similar labels or features. This assumption turns out to be reasonable in settings such as social (McPherson et al., 2001), regional planning (Gerber et al., 2013), and citation (Ciotti et al., 2016) networks. However, heterophilous graphs violate this assumption leading to sub-optimal performance (Zhu et al., 2020; 2021; Chien et al., 2021; Lim et al., 2021; Wang et al., 2023), owing to *oversmoothing* (Li et al., 2018) resulting from flattening of high-frequency information (Wu et al., 2023) by message-passing schemes.

We shed light on this issue with the standard QM9 data in Fig. 1. A conceptual way to characterize homophily is by examining the neighbours of each node. A fully *homophilous* molecule only has links between atoms of the same type (right), while a *heterophilous* molecule has links between different types (left). We observe that a major fraction of molecules in QM9 have scores in the range [0.4, 0.8].

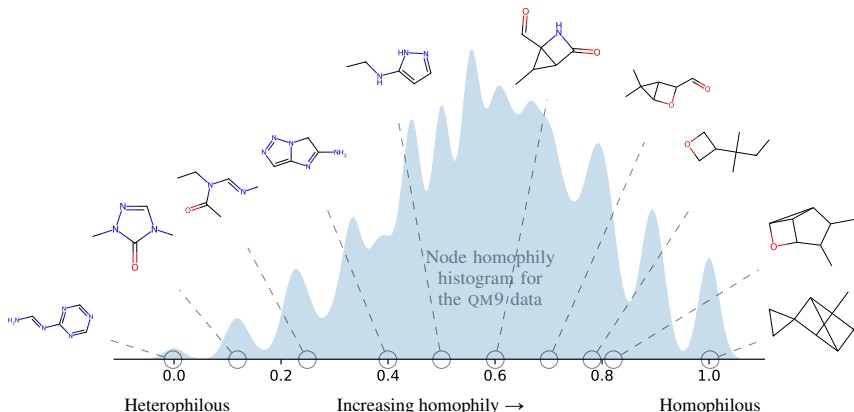

Figure 1: A simple way to characterize homophily is by studying the links of each node. A fully *homophilous* molecule only has links between atoms of the same type, while a *heterophilous* molecule has links between different types. Rather than counting links, HTFlow utilizes multiple interactive flows to estimate the propensity of a link to be homophilic/heterophilic in a given molecular context.

However, in practice, simply counting atom types is not expressive enough. Instead, the heterophily typically stems from more intricate properties of the molecules which need to be learned from data. We introduce HTFlows to carefully address and utilize the heterophily present in molecular data during generative modelling.

**Our contributions** In this paper, we introduce a novel framework for flow-based graph generation, likely the first molecular generation model that directly accounts for data heterophily. The proposed model comprises several interactive flows, designed to learn graph structures and node features across varying degrees of homophily and heterophily. Our key contributions are summarized below:

- **(Conceptual and technical)** we motivate the relevance of heterophily in molecular contexts, and propose a generative framework that encodes homophily/heterophily patterns;

- **(Methodological)** we design an invertible model with three co-evolving flows: a central flow interacts with heterophilous and homophilous flows to learn nuanced representations;

- **(Empirical)** we demonstrate the benefits of our method by benchmarking molecule generation on the QM9 and ZINC-250K data sets, evaluating with an extensive set of 14 different chemoinformatics metrics to analyze the actual chemical properties of the generated data.

Notable advantages of our model include achieving high validity without the need for additional validity checks in random generation experiments and successful optimization of target chemical properties in molecular searches. We now proceed to reviewing relevant related works.

## 2 RELATED WORK

**Molecule representation and generation** Early works in molecule generation (*e.g.*, Kusner et al., 2017; Guimaraes et al., 2017; Gómez-Bombarelli et al., 2018; Dai et al., 2018) primarily used sequence models to encode the SMILES strings (Weininger et al., 1989). Graphs afford more flexible modeling of interactions, so the field has gravitated towards representing molecules as (geometric) graphs and using powerful graph encoders, *e.g.*, based on graph neural networks (GNNs).

Variational autoencoders (VAEs, Kingma & Welling, 2014) provided a toolset for molecule generation with an encoder-decoder architecture, affording a latent encoding that can be optimized to search for molecules with specific properties. A prominent work, JT-VAE, showed benefits of viewing graphs as tree-like substructures obtained by including rings, in addition to the usual atoms labels, as part of the vocabulary (Jin et al., 2018). Other models such as Graph Convolutional Policy Network (You et al., 2018) and MolecularRNN (Popova et al., 2019; Shi et al., 2019; Luo et al., 2021) add atoms/bonds sequentially, and rely on rejection schemes to ensure the validity of the generated

molecules. Generative Adversarial Networks (GANs, Goodfellow et al., 2014) introduced added flexibility, as demonstrated by works such as De Cao & Kipf (2018) and You et al. (2018).

**Flow-based models** Normalizing flows enable exact likelihood estimation (see Papamakarios et al., 2021), so have recently gained prominence in the context of molecule generation (Kaushalya et al., 2019; Luo et al., 2021; Shi et al., 2019; Zang & Wang, 2020; Verma et al., 2022). These models learn invertible transformations to map data from a simpler base distribution to a more complex distribution over molecules. GraphAF (Shi et al., 2019) and GraphDF (Luo et al., 2021) keep the traditional sequential generation process, with GraphDF constraining the latent variables to be discrete. MoFlow (Zang & Wang, 2020) leverages a GLOW model (Kingma & Dhariwal, 2018) for structure generation with a conditional flow for assigning atom types.

More recently, there has been a shift towards incorporating prior knowledge and stronger inductive biases into deep learning models for molecule generation, thus allowing for more nuanced and accurate representations. ModFlow (Verma et al., 2022) builds a continuous normalizing flow with graph neural ODEs (Poli et al., 2019) assuming molecular structure is available, and use an E(3)-equivariant GNN (Satorras et al., 2021) to account for rotational and translational symmetries. EDM (Hoogeboom et al., 2022) generates molecules in 3D space through an equivariant diffusion model (Sohl-Dickstein et al., 2015; Song et al., 2021; Austin et al., 2021; Vignac et al., 2022) on the atom coordinates and categorical types. This relates to ongoing interest in guiding the generative process by controlling the inductive biases of the model. Such structure is perhaps more apparent in image generation (*e.g.*, Rissanen et al., 2023; Hoogeboom & Salimans, 2023), while in molecule modelling the prior knowledge needs to be included in more subtle ways, such as in the form of heterophily.

**Heterophily** Many previous studies analyze how heterophily influences GNN performance and design new methods to mitigate it (Zhu et al., 2020; Liu et al., 2021a; Yan et al., 2022; Ma et al., 2021). Some studies demonstrate deeper insights about how heterophily affects model expressiveness (Ma et al., 2021; Luan et al., 2022; Mao et al., 2023; Luan et al., 2023). However, most of these papers focus on node classification. However, molecular generation requires models to learn the data distribution by distinguishable graph embeddings. Heterophilic graphs lose distinguishability more from message-passing layers. We now address this issue with HTFlows.

## 3 HETEROPHILOUS TRIPLE FLOWS

We propose a graph generative model leveraging normalizing flows and heterophily features in graph data. Our model is split into two main components: the bond flow and the atom flow. The bond flow focuses on learning the molecular structure, while the atom flow assigns specific atomic details to this topology.

### 3.1 PREREQUISITES: NORMALIZING FLOWS WITH AFFINE COUPLING LAYERS

**Normalizing flows** offer a methodical approach to transform a simple distribution (like a Gaussian) into a complex one, matching the distribution of the target data. This is achieved by applying a chain of reversible and bijective transformations for distribution learning (Dinh et al., 2014). Given a flow $f = f_T \circ \cdots \circ f_1$, we initialize from a target distribution $z_0 \sim p_z$. The flow then undergoes a series of transformations to reach a Gaussian distribution $z_T \sim \mathrm{N}(\mu, \sigma^2)$ through invertible functions: $z_i = f_i(z_{i-1}), i = 1, 2, \ldots, T$. The goal of normalizing flows is to minimize the difference between the learned distribution and the target distribution. This is typically quantified using the negative log-likelihood of the data. The flow learns the

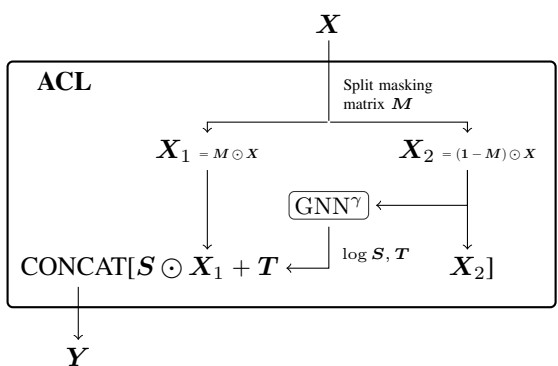

Figure 2: The affine coupling layer. The coupling is defined through a GNN and depends on the nature ($\gamma \in \{\mathrm{hom.}, \mathrm{cen.}, \mathrm{het.}\}$) of the flow (see Sec. 3.2).

target distribution by minimizing the negative log-likelihood:

$$\mathcal{L} = -\log p_{\boldsymbol{z}}(\boldsymbol{z}_0) = -\log \mathrm{N}(\boldsymbol{z}_T \mid \mu, \sigma^2) - \log \det |\partial f / \partial \boldsymbol{z}_0|. \tag{1}$$

The power of normalizing flows lies in their bijectiveness. Each transformation is both reversible and maintains the 'volume' of the data distribution. This ensures that no information from the data is lost during these transformations. Thus, the transformed distribution can be 'pulled back' to the original space using the inverse of the transformation functions, providing a bridge between the simple Gaussian and the intricate target distribution. For this to work, the flow needs to be reversible, which we get back to in Sec. 3.5.

**Affine coupling layers** (ACLs) introduce reversible transformations to normalizing flows, ensuring efficient computation of the log-determinant of the Jacobian (Kingma & Dhariwal, 2018). Typically, the affine coupling layer, denoted by $\mathrm{ACL}^{(f,\boldsymbol{M})}$, contains a binary masking matrix $\boldsymbol{M} \in \{0,1\}^{m \times n}$ and **coupling function** $f$ which determines the affine transformation parameters . Given an input $\boldsymbol{X} \in \mathbb{R}^{m \times n}$, the input is split into $\boldsymbol{X}_1 = \boldsymbol{M} \odot \boldsymbol{X}$ and $\boldsymbol{X}_2 = (1 - \boldsymbol{M}) \odot \boldsymbol{X}$ by masking, where '$\odot$' denotes the element-wise Hadamard product. Here, $\boldsymbol{X}_1$ is the masked input that will undergo the transformation, and $\boldsymbol{X}_2$ is the part that provides parameters for this transformation via the coupling function and keeps invariant insdide the ACLs. The output is the concatenation of the transformed part and the fixed part as visualized in Fig. 2:

$$\mathrm{ACL}^{(f,\boldsymbol{M})}(\boldsymbol{X}) = \boldsymbol{M} \odot (\boldsymbol{S} \odot \boldsymbol{X}_1 + \boldsymbol{T}) + (1 - \boldsymbol{M}) \odot \boldsymbol{X}_2 \quad \text{such that} \quad \log \boldsymbol{S}, \boldsymbol{T} = f(\boldsymbol{X}_2). \tag{2}$$

The binary masking eensures that only part of the input is transformed, allowing the model to retain certain features while altering others, enabling the flow to capture intricate data distribution characteristics. This is key for enabling heterophily in the next sections.

## 3.2 HETEROPHILIOUS MESSAGE PASSING

**Graph Neural Networks** (GNNs) have emerged as a potent paradigm for learning from graph-structured data, where the challenges include diverse graph sizes and varying structures (Kipf & Welling, 2017; Veličković et al., 2018; Xu et al., 2019; Garg et al., 2020). Consider a graph $G = (\mathcal{V}, \mathcal{E})$ with nodes $\mathcal{V}$ and edges $\mathcal{E}$. For these nodes and edges, we denote the corresponding node features as $\boldsymbol{X} = \{\boldsymbol{x}_v \in \mathbb{R}^{n_v} \mid v \in \mathcal{V}\}$ and edge features as $\boldsymbol{E} = \{\boldsymbol{e}_{uv} \in \mathbb{R}^{n_e} \mid u, v \in \mathcal{E}\}$. For each node $v \in \mathcal{V}$, its embedding at the $k^{\text{th}}$ layer is represented as $\boldsymbol{h}_v^{(k)}$. These embeddings evolve through a sequence of transformations across $K$-deep GNN, by the message passing scheme (Hamilton, 2020):

$$\boldsymbol{m}_{uv}^{(k)} = \mathrm{MESSAGE}_{uv}^{(k)}\left(\boldsymbol{h}_u^{(k)}, e_{uv}\right), \qquad u \in \mathcal{N}(v), \quad k = 0, 1, \dots, K, \tag{3}$$

$$\boldsymbol{h}_v^{(k+1)} = \mathrm{UPDATE}^{(k)}\left(\boldsymbol{h}_v^{(k)}, \boldsymbol{m}_{\mathcal{N}(v)}^{(k)}\right), \qquad k = 0, 1, \dots, K. \tag{4}$$

Here $\mathcal{N}(v)$ denotes the neighbours set of node $v$. Both $\mathrm{UPDATE}^{(k)}$ and $\mathrm{MESSAGE}_{uv}^{(k)}$ are arbitrary differentiable functions. The set $\boldsymbol{m}_{\mathcal{N}(v)}^{(k)} = \{\boldsymbol{m}_{uv}^{(k)} \mid u \in \mathcal{N}(v)\}$ aggregates messages from all neighbours of $v$. Importantly, the function $\mathrm{UPDATE}^{(k)}$ needs to be permutation invariant on this message set $\boldsymbol{m}_{\mathcal{N}(v)}^{(k)}$ (e.g., by operations like summation or taking the maximum). However, a naïve aggregation strategy will mix different messages and leads to the 'oversmoothing' problem.

**Heterophilious GNNs** Our method HTFlows encodes the heterophily assumption into the message passing sheme of the GNN. We denote the $\mathrm{GNN}^\gamma$ with heterophilious message passing scheme with an indicator $\gamma \in \{\mathrm{cen., hom., het.}\}$ depending on the scheme being employed. These indicators specify the preference of the GNNs: whether they lean towards homophily (hom.), centrality (cen.), or heterophily (het.).

Referring to Eq. (4), the messages undergo a preprocessing step before being sent forward to the subsequent layer. This is given by:

$$\boldsymbol{m}_{\mathcal{N}(v)}^{(k)} = \{\alpha_{uv}^{(k)} \boldsymbol{m}_{uv}^{(k)} \mid u \in \mathcal{N}(v)\}, \tag{5}$$

where

$$\alpha_{uv}^{(k)} = \begin{cases} \mathcal{H}(u,v), & \text{if } \gamma = \mathrm{hom.} \\ 1, & \text{if } \gamma = \mathrm{cen.} \\ 1 - \mathcal{H}(u,v), & \text{if } \gamma = \mathrm{het.} \end{cases} \tag{6}$$

where $\mathcal{H}$ denotes the homophily of the node (Pei et al., 2019). Yet, instead of traditional labels, in this context, the model aims to learn embeddings. Thus, in practice, we define the homophily or *attraction to similarity* between embeddings as the cosine similarity $\mathcal{H}(u, v) \triangleq S_{\cos}(\boldsymbol{h}_u^{(k)}, \boldsymbol{h}_v^{(k)})$ at the relevant layer.

### 3.3 HETEROPHILOUS TRAINING PROCESS

Given a molecule represented as a graph $G = (\boldsymbol{X}, \boldsymbol{B})$, the atom features are denoted by $\boldsymbol{X} \in \mathbb{R}^{n \times n_a}$ and the bond features by $\boldsymbol{B} \in \mathbb{R}^{n \times n \times n_b}$. The terms $n_a$ and $n_b$ represent the number of atom types and bond types, respectively. Specifically, $(\boldsymbol{X})_i$ denotes the one-hot encoded type of the $i^{\text{th}}$ atom present in molecule $G$. Similarly, $(\boldsymbol{B})_{ij}$ denotes the one-hot encoding of the specific chemical bond between the $i^{\text{th}}$ and $j^{\text{th}}$ atom in $G$. Our model HTFlows maps the molecule $G$ to embeddings $\boldsymbol{h}^{(a)}$ and $\boldsymbol{h}^{(b)}$ from the Gaussian distributions:

$$\boldsymbol{h}^{(a)} \sim p_a = \mathrm{N}(\mu_a, \sigma_a^2), \quad \boldsymbol{h}^{(b)} \sim p_b = \mathrm{N}(\mu_b, \sigma_b^2). \quad (7)$$

**Bond flow** The bond flow represented by $f_b = \mathrm{ACL}_{k_b}^b \circ \cdots \circ \mathrm{ACL}_1^b$ consists of a series of affine coupling layers with simple convolutional networks (CNNs) as coupling function: $\mathrm{ACL}_i^b = \mathrm{ACL}^{(\mathrm{CNN}_i, \boldsymbol{M}_i^b)}$, $i = 1, 2, \ldots, k_b$, where $k_b$ denotes the number of layers and masking $(\boldsymbol{M}_i^b)_{jk} = \mathbf{1}_{[2k/n_b] \equiv i(2)}$. Then bond embeddings $\boldsymbol{h}^{(b)} = \boldsymbol{B}_{k_b} = f_b(\boldsymbol{B}_0)$ emerge from the bond tensor $\boldsymbol{B}_0 = \boldsymbol{B}$:

$$\boldsymbol{B}_i = \mathrm{ACL}_i^b(\boldsymbol{B}_{i-1}), \quad i = 1, 2, \ldots, k_b. \quad (8)$$

**Heterophilous atom flow** The atom flow $f_a$ contains three dependent normalizing flows of depth $k_a$. They are the central, homophilic, and heterophilic flows, associated with specific indicators labelled as $\Gamma = \{\text{cen.}, \text{hom.}, \text{het.}\}$. The corresponding affine coupling layers are built with heterophilious GNNs defined in Sec. 3.2 as coupling functions and masking $\boldsymbol{M}_i \in \{0, 1\}^{n \times n \times n_b}$

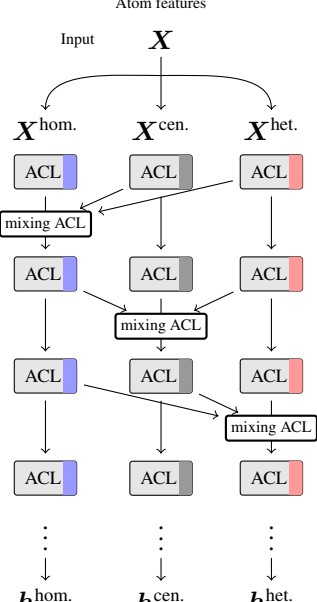

Atom features

Input $\boldsymbol{X}$

$\boldsymbol{X}^{\text{hom.}}$ $\boldsymbol{X}^{\text{cen.}}$ $\boldsymbol{X}^{\text{het.}}$

$\boldsymbol{h}^{\text{hom.}}$ $\boldsymbol{h}^{\text{cen.}}$ $\boldsymbol{h}^{\text{het.}}$

Figure 3: Heterophilous atom flow structure of HTFlows. The color of the ACL block refers to the indicators of GNN coupling functions: hom., cen., het.

$$\mathrm{ACL}_{i,\gamma}^a = \mathrm{ACL}^{(\mathrm{GNN}_i^\gamma, \boldsymbol{M}_i)}, \quad i = 1, 2, \ldots, k_a, \quad \gamma \in \Gamma. \quad (9)$$

where $(\boldsymbol{M}_i)_{j,k,l} = \mathbf{1}_{j \equiv i(n_a)}$. All GNNs in this context derive their graph topology $(\mathcal{E}, \boldsymbol{E})$ from the bond tensor $\boldsymbol{B}$. The embeddings are initialized by the atom features: $\boldsymbol{X}_0^\gamma = \boldsymbol{X}, \gamma \in \Gamma$. With each layer, the embeddings undergo an update through the coupling layers:

$$\bar{\boldsymbol{X}}_i^\gamma = \mathrm{ACL}_{i,\gamma}^a\left(\boldsymbol{X}_{i-1}^\gamma \mid \boldsymbol{B}\right), \quad i = 1, 2, \ldots, k_a, \quad \gamma \in \Gamma. \quad (10)$$

Instead of constructing three separate flows, another sequence of 'mixing' affine coupling layers is introduced: $\mathrm{ACL}_i^{\text{mix.}} = \mathrm{ACL}^{(\mathrm{MLP}_i, \boldsymbol{M}_i^{\text{mix.}})}$ with MLP coupling functions. These layers serve the purpose of facilitating interactions between flows. By modulating the mask matrix $\boldsymbol{M}_i^{\text{mix.}} \in \{0, 1\}^{n \times 3n_a}$, the three flows engage in iterative interactions:

$$\boldsymbol{h}^{(a)} = \boldsymbol{h}_{k_a}^{(a)} = f_a(\boldsymbol{h}_0^{(a)} \mid \boldsymbol{B}), \quad \boldsymbol{h}_i^{(a)} = \mathrm{ACL}_i^{\text{mix.}}\left(\bar{\boldsymbol{h}}_i^{(a)}\right), \quad i = 1, 2, \ldots, k_a, \quad (11)$$

where the embeddings are concatenated from the three flows as $\boldsymbol{h}_i^{(a)} = \mathrm{concat}\left[\boldsymbol{X}_i^{\text{cen.}}, \boldsymbol{X}_i^{\text{hom.}}, \boldsymbol{X}_i^{\text{het.}}\right]$ and $\bar{\boldsymbol{h}}_i^{(a)} = \mathrm{concat}\left[\bar{\boldsymbol{X}}_i^{\text{cen.}}, \bar{\boldsymbol{X}}_i^{\text{hom.}}, \bar{\boldsymbol{X}}_i^{\text{het.}}\right]$, and the mask matrix $(\boldsymbol{M}_i^{\text{mix.}})_{jk} = \mathbf{1}_{[k/n_a] \equiv i(3)}$. A visual representation of the entire structure of the HTFlows model can be found in Fig. 3. For better undersatnding, we provide example reconstructions from intermediate layers in Fig. 4.

**Loss** The loss function combines the negative log-likelihoods (NLLs) from both the atom and bond flows: $\mathcal{L} = \mathcal{L}_a + \mathcal{L}_b$. Each NLL could be decomposed as shown in Eq. (1):

$$\mathcal{L}_b = -\log p_b\left(\boldsymbol{h}^{(b)}\right) - \log \det\left(\left|\frac{\partial \boldsymbol{h}^{(b)}}{\partial \boldsymbol{B}}\right|\right) = -\log p\left(\boldsymbol{h}^{(b)}\right) - \sum_{i=1}^{k_b} \log \det\left(\left|\frac{\partial \mathrm{ACL}_i^b(\boldsymbol{B}_{i-1})}{\partial \boldsymbol{B}_{i-1}}\right|\right). \quad (12)$$

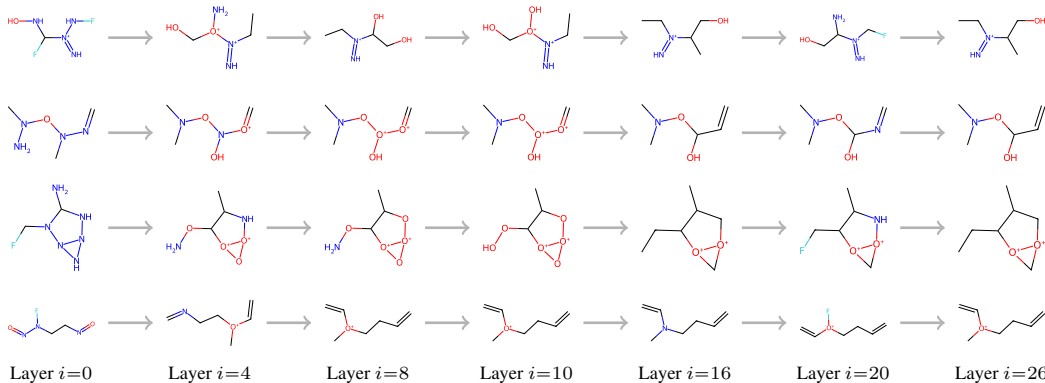

Figure 4: **Step-by-step generation (QM9).** Snapshots of reconstructed molecules when fixing the bond model and collecting node embeddings of the intermediate layers $i$.

Similarly, the loss $\mathcal{L}_a$ for the heterophilous atom flow can be constructed as:

$$
\mathcal{L}_a = -\log p\left(\boldsymbol{h}_{k_a}^{(a)}\right) - \log \det\left(\left|\frac{\partial \boldsymbol{h}^{(a)}}{\partial \boldsymbol{X}}\right|\right)
$$

$$
= -\log p\left(\boldsymbol{h}_{k_a}^{(a)}\right) - \sum_{i=1}^{k_a}\left[\log \det\left(\left|\frac{\partial \mathrm{ACL}_i^{\mathrm{mix.}}\left(\bar{\boldsymbol{h}}_i^{(a)}\right)}{\partial \bar{\boldsymbol{h}}_i^{(a)}}\right|\right) - \sum_{\gamma \in \Gamma}\log \det\left(\left|\frac{\partial \mathrm{ACL}_{i,\gamma}^a\left(\boldsymbol{X}_{i-1}^\gamma\right)}{\partial \boldsymbol{X}_{i-1}^\gamma}\right|\right)\right].
$$
(13)

### 3.4 GENERATION PROCESS

Given a trained HTFlows model, with established atom flow $f_{a*}$ and bond flow $f_{b*}$, the procedure for generating molecules is described as follows.

1. **Sampling Embeddings:** Start by randomly sampling embeddings $\boldsymbol{h}^{(a)}$ and $\boldsymbol{h}^{(b)}$ from a Gaussian distribution as expressed in Eq. (7).

2. **Obtaining the Bond Tensor:** The bond tensor $\boldsymbol{B}$ can be derived by applying the inverse of the bond flow $f_{b*}$ to the sampled embedding $\boldsymbol{h}^{(b)}$. This is given as

$$
\boldsymbol{B} = f_{b*}^{-1}(\boldsymbol{h}^{(b)}) = \left(\mathrm{ACL}_{1*}^b\right)^{-1} \circ \cdots \circ \left(\mathrm{ACL}_{k_b*}^b\right)^{-1}(\boldsymbol{h}^{(b)}).
$$
(14)

3. **Recovering Graph Topology:** From the bond tensor $\boldsymbol{B}$, the graph topology $(\mathcal{E}, \boldsymbol{E})$ can be deduced. This topology is essential for the GNN-based affine coupling layers (ACLs) within the atom flow $f_a$.

4. **Generating Node Features:** With the bond tensor in place, node features can be produced by applying the inverse of the atom flow $f_{a*}$ to the sampled atom embedding $\boldsymbol{h}^{(a)}$. This is given as

$$
\boldsymbol{X} = f_{a*}^{-1}(\boldsymbol{h}^{(a)} \mid \boldsymbol{B}).
$$
(15)

5. **Molecule Recovery:** Finally, a molecule, represented as $G$, can be reconstructed using the generated atom features $\boldsymbol{X}$ and bond tensor $\boldsymbol{B}$ from random embeddings $[\boldsymbol{h}^{(a)}, \boldsymbol{h}^{(b)}]$.

### 3.5 REVERSIBILITY OF THE HETEROPHILOUS TRIPLE FLOWS

To ensure that the molecular embeddings and transformations produced by HTFlows can be inverted back, it is crucial to understand the reversibility of the processes. Both the atom and bond models of HTFlows rely on ACL blocks. As introduced in Sec. 3.1, these blocks are inherently reversible. This means they can forward process the input to produce an output and can also take that output to revert it back to the original input without loss of information. Besides the use of ACL blocks,

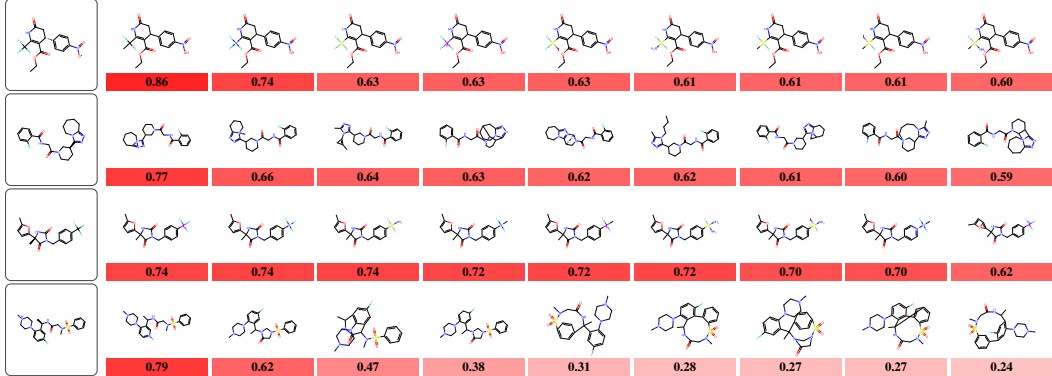

Figure 5: **Structured latent-space exploration (ZINC-250K).** Nearest neighbour search in the latent space with the seed molecules on the left and neigbours with the Tanimoto similarity (1 ▇▇▇▇ 0) given for each molecule. For results on QM9, see Fig. A8 in the appendix.

the operations used within the model primarily leverage simple concatenation or splitting. These operations are straightforward and do not affect the overall reversibility of the processes. Given that the individual components (both atom and bond flows) are reversible and the operations performed on the data are straightforward, it is apparent that HTFlows as a whole is reversible. A formal proof on reversibility of ACL blocks and HTFlows is provided in App. B.

## 4 EXPERIMENTS

We demonstrate our model in a variety of common benchmarks tasks for molecule generation and modelling. First, provide an illustrative example of latent space exploration around seed molecules. Second, we provide results for molecule generation with benchmarks on a wide range of chemoinformatics metrics. Finally, we provide results for molecular property optimization.

**Implementation** The models were implemented in PyTorch (Paszke et al., 2019) and PyTorch Geometric (Fey & Lenssen, 2019). In HTFlows, we used GNNs with 4 layers and flows that were $k_a = 27$(QM9) and $k_a = 38$(ZINC-250K) and $k_b = 10$ deep. We trained our models with the AdamW optimizer (Loshchilov & Hutter, 2019) for 500 epochs, with batch size 256 and learning rate 0.001. The final model selection was based on score comparison on a hold-out validation set. We select the best-performin model by the FCD score as suggested in Polykovskiy et al. (2020). Our models are trained on a cluster equipped with NVIDIA A100 GPUs. The training time for single models were 24 hours (QM9) and 56 hours (ZINC-250K).

**Chemoinformatics metrics** We compare methods through an extensive set of chemoinformatics metrics that perform both sanity checks (validity, uniqueness, and novelty) on the generated molecule corpus and quantify properties of the molecules: neighbour (SNN), fragment (Frag), and scaffold (scaf) similarity, internal diversity (IntDiv$_1$ and IntDiv$_2$), and Fréchet ChemNet distance (FCD). We also show score histograms for solubility (logP), syntetic accessibility (SA), drug-likeness (QED), and molecular weight. For computing the metrics, we use the MOSES benchamrking platform (Polykovskiy et al., 2020) and the RDKit open-source cheminformatics software (Landrum et al., 2013). The 'data' row in metrics is based on randomly sampled (1000 mols) for 10 times from data set. When we calculate the metrics we simulate 1000 molecules for 10 times and compare them to a hold-out reference set (20% of data, other 80% is used for training). Full details on the 14 metrics we use are included in App. C.

**Data sets** We consider two common molecule data sets: QM9 and ZINC-250K. The QM9 data set (Ramakrishnan et al., 2014) comprises $\sim 134$k stable small organic molecules composed of atoms from the set {C, H, O, N, F}. These molecules have been processed into their kekulized forms with hydrogens removed using the RDkit software (Landrum et al., 2013). The ZINC-250K (Irwin et al., 2012) data contains $\sim 250$k drug-like molecules, each with up to 38 atoms of 9 different types. Despite still relatively small (molecular weights ranging from 100 to 500), the molecules in the ZINC-250K data set are larger and more complicated than those in QM9.

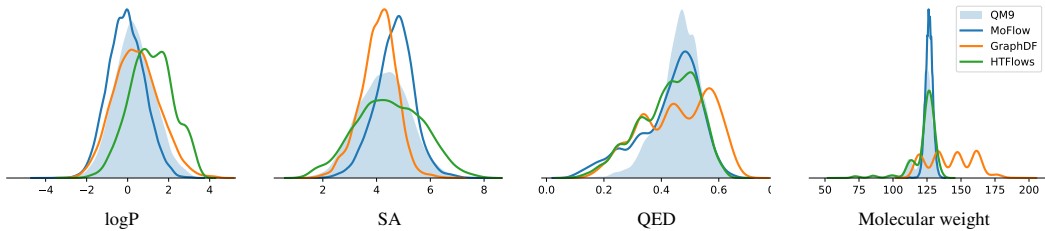

Figure 6: Chemoinformatics statistics for data (QM9) and generated molecules from HTFlows (ours), MoFlow, and GraphDF. We report histograms for the Octanol-water partition coefficient (logP), synthetic accessibility score (SA), quantitative estimation of drug-likeness (QED), and molecular weight.

Table 1: Chemoinformatics summary statistics for random generation on the QM9 data set. Full listing of all 14 metrics in Table A4. HTFlows performs well on all these summary metrics.

| | FCD ↓ | SNN ↑ | Frag ↑ | Scaf ↑ | IntDiv$_1$ ↑ | IntDiv$_2$ ↑ |
|---|---|---|---|---|---|---|
| Data (QM9) | 0.40 | 0.54 | 0.94 | 0.76 | 0.92 | 0.90 |
| GraphDF | 10.76 | 0.35 | 0.61 | 0.09 | 0.87 | 0.86 |
| MoFlow | 7.48 | 0.33 | 0.60 | 0.04 | **0.92** | **0.90** |
| HTFlows | **5.63** | **0.36** | **0.71** | **0.23** | 0.92 | 0.90 |

**Visualizing the continuous latent space**   Similar to Zang & Wang (2020), we examine the learned latent space of our method on both QM9 and ZINC-250K. Results for ZINC-250K are presented in Fig. 5 and QM9 in the appendix (Fig. A8). Qualitatively, we note that latent space appears smooth and the molecules near the seed molecule resemble the input and have high Tanimoto similarity (Rogers & Hahn, 2010).

## 4.1   Molecule generation

**Baselines**   For random generation, we include baseline results for models that have pre-trained models available. We need access to the trained models, because few papers report chemoinformatics metrics beyond trivial sanity checks (validity, uniqueness, and novelty) that tend to be high (90%–100%) for most models. We compare to GraphDF (Luo et al., 2021) and MoFlow (Zang & Wang, 2020) which are current state-of-the-art (see Verma et al., 2022).

**Results on QM9**   For the QM9 data set, the main chemoinformatic summary statistics are given in Table 1 and the descriptive distributions in Fig. 6. The full listing of all 14 metrics is provided in Table A4 in the appendix. From Table 1, HTFlows achieves the lowest FCD, and achives highest (or on-par values with MoFlow) on SNN, Frag, and diversity. From the extended results in Table A4, it is clear that each model has its strengths, and the choice might depend on the specific requirements of a task. If one is looking for a model that produces a broad range of diverse molecules, HTFlows stands out as preferable.

**Results on ZINC-250K**   For the ZINC-250K data set, the main chemoinformatic summary statistics are given in Table 2 and the descriptive distributions in Fig. 7. The full listing of all 14 metrics are provided in Table A5 in the appendix. While its performance on the ZINC-250K data set exhibits some variation, HTFlows still achieves the best internal diversity (IntDiv$_1$ and IntDiv$_2$) and has the most favorable molecular weight distribution. Notably, its validity score is lower than MoFlow, indicating some challenges in generating completely valid molecules in this context. Overall, HTFlow emerges as a robust and versatile molecular generation model, adept at balancing fidelity, diversity, and molecular properties.

## 4.2   Property optimizaiton

In the property optimization task, models show their capability in finding novel molecules that optimize specific chemical properties not present in the training data set: a critical component for drug discovery. For our study, we focused on maximizing the QED property. We trained HTFlows on

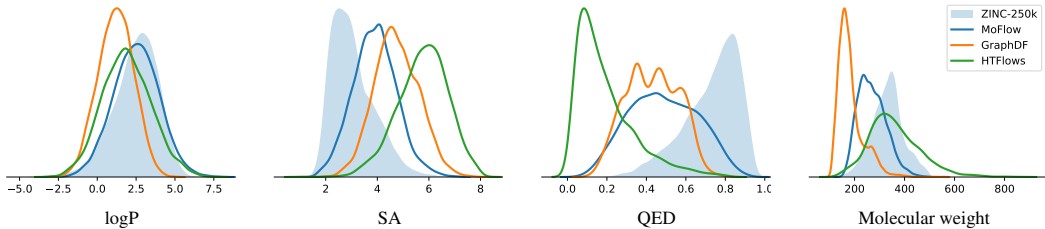

Figure 7: Chemoinformatics statistics for data (ZINC-250K) and generated molecules from HTFlows (ours), MoFlow, and GraphDF. Histograms for the Octanol-water partition coefficient (logP), synthetic accessibility score (SA), quantitative estimation of drug-likeness (QED), and molecular weight.

Table 2: Chemoinformatics summary statistics for random generation on the ZINC-250K data set. Full listing of all 14 metrics in Table A5. HTFlows performs well especially on diversity metrics.

| | FCD ↓ | SNN ↑ | Frag ↑ | Scaf ↑ | IntDiv$_1$ ↑ | IntDiv$_2$ ↑ |
|---|---|---|---|---|---|---|
| Data (ZINC-250K) | 1.44 | 0.51 | 1.00 | 0.28 | 0.87 | 0.86 |
| GraphDF | 34.30 | 0.23 | 0.35 | 0.00 | 0.88 | 0.87 |
| MoFlow | **22.65** | **0.29** | **0.81** | **0.01** | 0.88 | 0.86 |
| HTFlows | 27.90 | 0.22 | 0.57 | 0.00 | **0.90** | **0.88** |

ZINC-250K and evaluated its performance against other state-of-the-art models (Verma et al., 2022; Luo et al., 2021; Zang & Wang, 2020; Shi et al., 2019; Jin et al., 2018; You et al., 2018). The results, given in Table 3, show that the top three novel molecule candidates identified by HTFlows (that are not part of the ZINC-250K data set), exhibit QED values on par with those from ZINC-250K or other state-of-the-art methods. For details of the property optimization strategy and the top three molecules, see App. D.2.

## 5 DISCUSSION AND CONCLUSIONS

We have presented HTFlows, a novel approach to molecular generation by emphasizing heterophily patterns, countering the traditional oversmoothing vulnerability seen in existing graph neural network methodologies. By leveraging multiple interactive flows to discern (dis-)similarities between molecular entities, our method offers a more versatile representation of the intricate balance between molecular affinities and repulsions. The experiment results show HTFlows' ability to consistently generate molecules with high fidelity, diversity, and desired properties, marking it as a promising tool in the field of chemoinformatics and molecular design.

Based on the experiment results, it is noteworthy to draw parallels and distinctions between our model and MoFlow (Zang & Wang, 2020). While there are over-

Table 3: Performance on molecule property optimization in terms of the best QED scores, scores taken from the corresponding papers (JTVAE score from Luo et al., 2021; Verma et al., 2022).

| Method | 1st | 2nd | 3rd |
|---|---|---|---|
| Dat(ZINC-250K) | 0.948 | 0.948 | 0.948 |
| JTVAE | 0.925 | 0.911 | 0.910 |
| GCPN | **0.948** | 0.947 | 0.946 |
| GraphAF | **0.948** | **0.948** | 0.947 |
| GraphDF | **0.948** | **0.948** | **0.948** |
| MoFlow | **0.948** | **0.948** | **0.948** |
| ModFlow | **0.948** | **0.948** | 0.945 |
| HTFlows | **0.948** | **0.948** | **0.948** |

arching similarities, our approach introduces several enhancements. Foremost, our atom model incorporates a heterophilous message-passing scheme within the coupling layers of the GNNs, and employs multiple interactive flows for dynamic information exchange. MoFlow's implementation uses an additional dimension to represent non-existent nodes, which, in practice, reduces the GNNs to MLPs. Furthermore, the masking matrix in MoFlow's ACL layers filters information predicated on node order in each graph, inadvertently making the model susceptible to isomorphic transformations. In contrast, our HTFlows model allows flexible-sized input graphs, avoids message exchange from the non-existed nodes, and is permutation-invariant to isomorphism.

**Reproducibility statement** The code and trained models will be made available under the MIT License on GitHub upon acceptance.

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

# APPENDIX

This appendix is organized as follows. App. A presents details on heterophilious message passing in our model and compuational issues. App. B provides a formal proof to show that the proposed triple flow model is reversible. App. C summarizes and describes the metrics used in the experiments. App. D provides additional experiment results for molecule generation and the algorithm for property optimization together with detals on found candidate molecules.

# A  ALGORITHM DETAILS

## A.1  THE BEHAVIOR OF HETEROPHILIOUS GNN CONTROLLED BY $\gamma$

In the code implementation, we choose a single-layer Graph Attention Network (GAT) as the base, but change the scaling of the message collected from neighbours based on the homophily during message passing.

Given node embeddings $\boldsymbol{h}_u, \boldsymbol{h}_v$ of node $u, v$, there is the attention factor $\beta_{u,v}$ calculated based on a softmax of the attributes of nodes and edge feature $\boldsymbol{e}_{uv}$:

$$\beta_{u,v} = \frac{\exp\left(\mathrm{LeakyReLU}\left(\boldsymbol{a}^\top[\boldsymbol{\Theta}\boldsymbol{h}_u \,\|\, \boldsymbol{\Theta}\boldsymbol{h}_v \,\|\, \boldsymbol{\Theta}_e\mathbf{e}_{i,j}]\right)\right)}{\sum_{w\in\mathcal{N}(u)\cup\{u\}}\exp\left(\mathrm{LeakyReLU}\left(\boldsymbol{a}^\top[\boldsymbol{\Theta}\boldsymbol{h}_u \,\|\, \boldsymbol{\Theta}\boldsymbol{h}_w \,\|\, \boldsymbol{\Theta}_e\mathbf{e}_{v,w}]\right)\right)},$$

where $\theta = (\Theta, \Theta_e, \boldsymbol{a})$ are model parameters, Then the message collected from neighbours and updated to be

$$\boldsymbol{h}'_v = \alpha^\gamma_{v,v}\beta_{v,v}\boldsymbol{\Theta}\boldsymbol{h}_v + \sum_{u\in\mathcal{N}(v)} \alpha^\gamma_{u,v}\beta_{u,v}\boldsymbol{\Theta}\boldsymbol{h}_u,$$

where $\alpha^\gamma_{u,v}$ denotes the homophily factor, where

$$\alpha^\gamma_{u,v} = \begin{cases} 1, & \text{if } \gamma = \text{cen.} \\ \mathcal{H}(u,v), & \text{if } \gamma = \text{hom.} \\ 1 - \mathcal{H}(u,v), & \text{if } \gamma = \text{het.,} \end{cases}$$

where $\mathcal{H}(u,v) \triangleq S_{\cos}(\boldsymbol{h}_u^{(k)}, \boldsymbol{h}_v^{(k)})$ is the cosine similarity.

In conclusion, given the input $\boldsymbol{X} = [\boldsymbol{x}_v]_{v\in V}$, edge attributes and edge index contained inside the edge tensor $\mathbf{E}$, the GNN gets the output $\boldsymbol{X}' = [\boldsymbol{x}'_v]_{v\in V}$

$$\mathrm{GNN}^\gamma_\theta(\boldsymbol{X} \mid \mathbf{E}) = \boldsymbol{X}'.$$

## A.2  COMPUTATIONAL CONSIDERATIONS

For the convenience on the calculation of the log-likelihood, every transformation of variables needs the calculation of a Jacobian matrix (*i.e.*, $\partial \boldsymbol{Z}^{(l+1)}/\partial \boldsymbol{Z}^{(l)}$). So all the complicated modules (*e.g.*, GNNs, MLPs) are all built inside the coupling structure (part of input is updated by the scaling matrix $\boldsymbol{S}$, and transformation matrix $\boldsymbol{T}$ depends on the other part of input).

# B  PROOF OF REVERSIBILITY

## B.1  REVERSIBILITY OF THE ACL

**Set up**  Assume an ACL defined in Sec. 3.1 contains coupling function $f$ and masking matrix $\boldsymbol{M} \in \{0,1\}^{m\times n}$. Given input $\boldsymbol{X} \in \mathbb{R}^{m\times n}$, the output $\boldsymbol{Y}$ is calculated as

$$\boldsymbol{Y} = \mathrm{ACL}^{(f,\boldsymbol{M})}(\boldsymbol{X}) = \boldsymbol{M} \odot (\boldsymbol{S} \odot \boldsymbol{X}_1 + \boldsymbol{T}) + (\boldsymbol{1} - \boldsymbol{M})\boldsymbol{X}_2 \tag{16}$$

where $\log \boldsymbol{S}, \boldsymbol{T} = f(\boldsymbol{X}_2)$, and $\boldsymbol{X}_1, \boldsymbol{X}_2$ are the split from input by masking:

$$\boldsymbol{X}_1 = \boldsymbol{M} \odot \boldsymbol{X}, \quad \boldsymbol{X}_2 = (\boldsymbol{1} - \boldsymbol{M}) \odot \boldsymbol{X}.$$

We seek to recover $\boldsymbol{X}$ from the $f, \boldsymbol{M}$, and $\boldsymbol{Y}$.

**Reversibility from output to input** Since $M$ is binary, we get

$$M \odot M = M, \quad (1 - M) \odot (1 - M) = (1 - M),$$
$$M \odot (1 - M) = (1 - M) \odot M = 0$$

and

$$X = (M + (1 - M)) \odot X = M \odot X + (1 - M) \odot X = X_1 + X_2.$$

By splitting the output $Y$ to $Y_1, Y_2$ by masking matrix:

$$Y_1 = M \odot Y, \quad Y_2 = (1 - M) \odot Y.$$

Combining with Eq. (16), we know

$$
\begin{aligned}
Y_1 &= M \odot Y \\
&= M \odot (M \odot (S \odot X_1 + T) + (1 - M) \odot X_2) \\
&= M \odot (S \odot X_1 + T),
\end{aligned}
$$

and

$$
\begin{aligned}
Y_2 &= (1 - M) \odot Y \\
&= (1 - M) \odot (M \odot (S \odot X_1 + T) + (1 - M) \odot X_2) \\
&= (1 - M) \odot (M \odot (S \odot X_1 + T) + (1 - M) \odot (1 - M) \odot X) \\
&= (1 - M) \odot X = X_2.
\end{aligned}
$$

Now the $\log S, T = f(X_2) = Y_2$ are recovered by $Y$. Notice that

$$
\begin{aligned}
M \odot (Y_1 - T) \oslash S &= M \odot (M \odot (S \odot X_1 + T) - T) \oslash S \\
&= (M \odot S \odot X_1 + M \odot T - M \odot T) \oslash S \\
&= (M \odot S \odot X_1) \oslash S \\
&= M \odot X_1 \\
&= M \odot M \odot X \\
&= M \odot X \\
&= X_1 \quad \text{if} \quad (S)_{i,j} > 0, \quad \forall i, j,
\end{aligned}
$$

where '$\oslash$' denotes element-wise division. Since we define $S$ as the exponential of part of output from coupling function, the elements of $S$ are all strictly positive. Then

$$
\begin{aligned}
\left( \mathrm{ACL}^{(f, M)} \right)^{-1} (Y) &= X = X_1 + X_2 \\
&= M \odot (Y_1 - T) \oslash S + Y_2 \\
&= M \odot (M \odot Y - T) \oslash S + (1 - M) \odot Y.
\end{aligned}
\tag{17}
$$

where $\log S, T = f(X_2) = f((1 - M) \odot Y)$. Eq. (17) shows how the input is recovered from output, thus the ACL block is reversible.

## B.2 REVERSIBILITY OF THE BOND MODEL

For the bond model $f_b = \mathrm{ACL}^b_{k_b} \circ \cdots \circ \mathrm{ACL}^b_1$, and since each $\mathrm{ACL}^b_i$, $i = 1, \ldots, k_b$ is reversible, we can write $f_b^{-1} = \left( \mathrm{ACL}^b_1 \right)^{-1} \circ \cdots \circ \left( \mathrm{ACL}^b_{k_b} \right)^{-1}$, which the reverse function of $f_b$.

## B.3 REVERSIBILITY OF THE ATOM MODEL

For atom model $f_a$, which includes all $\{\mathrm{ACL}^{\mathrm{mix.}}_i, \mathrm{ACL}^a_{i,\gamma} | i = 1, \ldots, k_a, \gamma \in \Gamma\}$, we prove that each layer of $f_a$ which maps $h^{(a)}_{i-1}$ to $h^{(a)}_i$ is reversible.

For $i \in \{1, \ldots, k_a\}$, given $h^{(a)}_i = \mathrm{ACL}^{(\mathrm{mix.})}_i(\bar{h}^{(a)}_i)$, the $\bar{h}^{(a)}_i$ could be recovered by reversible $ACL^{(\mathrm{mix.})}_i$. Since

$$\bar{h}^{(a)}_i = \mathrm{concat}\left[ \bar{X}^{\mathrm{cen.}}_i, \bar{X}^{\mathrm{hom.}}_i, \bar{X}^{\mathrm{het.}}_i \right] \tag{18}$$

$$= \mathrm{concat}\left[ \mathrm{ACL}_{i,\mathrm{cen.}}(X^{\mathrm{cen.}}_{i-1}), \mathrm{ACL}_{i,\mathrm{hom.}}(X^{\mathrm{hom.}}_{i-1}), \mathrm{ACL}_{i,\mathrm{het.}}(X^{\mathrm{het.}}_{i-1}) \right], \tag{19}$$

then the $X^{\mathrm{cen.}}_{i-1}, X^{\mathrm{hom.}}_{i-1}, X^{\mathrm{het.}}_{i-1}$ can be recovered by reversible $\{ACL^a_{i,\gamma} \mid \gamma \in \Gamma\}$, thus $h^{(a)}_{i-1} = \mathrm{concat}\left[ X^{\mathrm{cen.}}_{i-1}, X^{\mathrm{hom.}}_{i-1}, X^{\mathrm{het.}}_{i-1} \right]$ is recovered.

## C    DESCRIPTION OF METRICS

For benchmarking, model selection, comparison, and explorative analysis, we use the following 14 metrics. The metrics are presented in detail in the work by Polykovskiy et al. (2020) that introduced the MOSES benchmarking platform. The metrics calculation makes heavy use of the RDKit open-source cheminformatics software (https://www.rdkit.org/). We briefly summarize the metrics below.

**Sanity check metrics**

1. **Validity** Fraction (in $[0, 1]$) of the molecules that produce valid SMILES representations. This is a sanity check for how well the model captures explicit chemical constraints such as proper valence. Higher values are better as a low value can indicate that the model does not capture properly chemical structure. We report numbers without *post hoc* validity corrections.

2. **Uniqueness** Fraction (in $[0, 1]$) of the molecules that are unique. This is a sanity check based on the SMILES string representation of the generated molecules. Higher values are better as a low value can indicate the model has collapsed and produces only a few typical molecules.

3. **Novelty** Fraction (in $[0, 1]$) of the generated molecules that are not present in the training set. Higher values are better as a low value can indicate overfitting to the training data set.

**Summary statistics**

4. **Similarity to a nearest neighbour (SNN)** The average Tanimoto similarity (Jaccard coefficient) in $[0, 1]$ between the generated molecules and their nearest neighbour in the reference data set. Higher is better: If the generated molecules are far from the reference set, similarity to the nearest neighbour will be low.

5. **Fragment similarity (Frag)** Measures similarity (in $[0, 1]$) of distributions of BRICS fragments (substructures) in the generated set vs. the original data set. If molecules in the two sets share many of the same fragments in similar proportions, the Frag metric will be close to 1 (higher better).

6. **Scaffold similarity (Scaf)** Measures similarity (in $[0, 1]$) of distributions of Bemis–Murcko scaffolds (molecule ring structures, linker fragments, and carbonyl groups) in the generated set vs. the original data set. This metric is calculated similarily as the Fragment similarity metric by counting substructure presence in the data, and they can be high even if the data sets do not contain the same molecules.

7. **Internal diversity (IntDiv$_1$)** Measure (in $[0, 1]$) of the chemical diversity within the generated set of molecules. Higher values are better and signal higher diversity in the generated set of moleculers. Low values can signal mode collapse.

8. **Internal diversity (IntDiv$_2$)** Measure (in $[0, 1]$) of the chemical diversity within the generated set of molecules. The interpretation is similar to IntDiv$_1$, but with stronger penalization of the Tanimoto similarity in calculating the diversity.

9. **Filters** This metric is specific to the MOSES benchmarking platrofm (see Polykovskiy et al., 2020). It gives the fraction (in $[0, 1]$) of generated molecules that passes filters applied during data set construction. In practice, these filters may filter out chemically valid molecules that have fragments that are not of interest in the MOSES data set (filtered with medicinal chemistry filters). Thus, this metric is not of primary interest for us, but gives a view on match with the MOSES data set.

10. **Fréchet ChemNet distance (FCD)** Analogous to the Frechét Inception Distance (FID) used in image generation, FCD compares feature distributions of real and generated molecules using a pre-trained model (ChemNet). Lower values are better.

**Descriptive distributions**

11. **Octanol-water partition coefficient (logP)** A logarithmic measure of the relationship between lipophilicity (fat solubility) and hydrophilicity (water solubility) of a set of molecules.

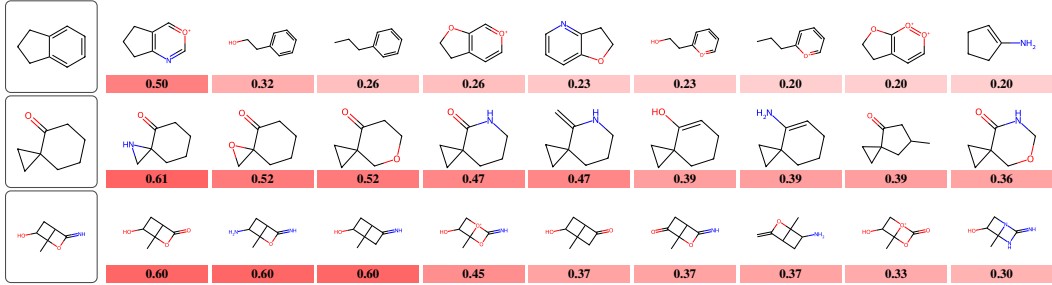

Figure A8: **Structured latent-space exploration (QM9).** Nearest neighbour search in the latent space with the seed molecule on the left and neigbours with the Tanimoto similarity (1 ▮▮▮▮ 0) given for each molecule.

For large values a substance is more soluble in fat-like solvents such as n-octanol, and for small values more soluble in water. We report both histograms of logP and a summary statistic in terms of the Wasserstein distance between the generated and reference distributions (smaller better).

12. **Synthetic accessibility score (SA)** A metric that estimates how easily a chemical molecule can be synthesized. It provides a quantitative value indicating the relative difficulty or ease of synthesizing a molecule, with a lower SA score suggesting that a molecule is more easily synthesized, and a higher score suggesting greater complexity or difficulty. We report both histograms of SA and a summary statistic in terms of the Wasserstein distance between the generated and reference distributions (smaller better).

13. **Quantitative estimation of drug-likeness (QED)** A metric designed to provide a quantitative measure of how 'drug-like' a molecule is. It essentially refers to the likelihood that a molecule possesses properties consistent with most known drugs, estimated based on a variety of molecular descriptors. We report both histograms of QED and a summary statistic in terms of the Wasserstein distance between the generated and reference distributions (smaller better).

14. **Molecular weight (Weight)** The sum of atomic weights in a molecule. We report both histograms of molecular weights and a summary statistic in terms of the Wasserstein distance between the generated and reference distributions (smaller better).

## D EXPERIMENT DETAILS

### D.1 FURTHER RESULTS

We provide further results for structured latent-space exploration (only ZINC-250K included in the main paper). Example explorations for QM9 are shown in Fig. A8.

We include full listings of all 14 metrics (description of metrics in App. C) considered in the random generation tasks for QM9 and ZINC-250K. The values are listed in Tables A4 and A5, respectively. Additionally, we also visualize the node homophily (in the 'neighbour-counting' sense as in Fig. 1) for both QM9 and ZINC-250K together with the estimated node homophily histograms (see Fig. A9) from the generation output from the different models. Even if our model, considers homophily/heterophily in learned embedding sense, the histograms show structure even for node homophily—though with an additional mode for strong heterophily, which shows for both HTFlows and MoFlow.

### D.2 PROPERTY OPTIMIZATION

**Algorithm** Given a pretrained HTFlows $f$, and training set $\mathcal{D}$ contains molecule and property label pairs $\{G, y\}$. Now we introduce an extra simple MLP $g_\theta$, trained on the dataset to be $g_{\theta*}$ by optimizing the parameters:

$$\theta^* = \arg\min_\theta \operatorname*{MSEloss.}_{(G,y)\in\mathcal{D}}(g_\theta(f(G)), y) \tag{20}$$

Table A4: Full benchmark metrics for random generation using QM9 (reporting mean±std).

|  | Validity ↑ | Uniqueness ↑ | Novelty ↑ | SNN ↑ | Frag ↑ | Scaf ↑ | IntDiv$_1$ ↑ |
|---|---|---|---|---|---|---|---|
| Data (QM9) | 1.00±0.00 | 1.00±0.00 | 0.62±0.02 | 0.54±0.00 | 0.94±0.01 | 0.76±0.03 | 0.92±0.00 |
| GraphDF | - | **1.00**±0.00 | 0.98±0.00 | 0.35±0.00 | 0.61±0.01 | 0.09±0.07 | 0.87±0.00 |
| MoFlow | **0.94**±0.01 | **1.00**±0.00 | **0.99**±0.00 | 0.33±0.00 | 0.60±0.03 | 0.04±0.03 | **0.92**±0.00 |
| HTFlows | 0.83±0.01 | **1.00**±0.00 | 0.95±0.01 | **0.36**±0.01 | **0.71**±0.04 | **0.23**±0.05 | **0.92**±0.00 |

|  | IntDiv$_2$ ↑ | Filters ↑ | FCD ↓ | logP ↓ | SA ↓ | QED ↓ | Weight ↓ |
|---|---|---|---|---|---|---|---|
| Data (QM9) | 0.90±0.00 | 0.64±0.02 | 0.40±0.02 | 0.04±0.01 | 0.03±0.01 | 0.00±0.00 | 0.32±0.08 |
| GraphDF | 0.86±0.00 | **0.69**±0.02 | 10.76±0.21 | **0.16**±0.03 | 0.27±0.02 | 0.05±0.00 | 19.72±0.54 |
| MoFlow | **0.90**±0.00 | 0.55±0.02 | 7.48±0.23 | 0.38±0.02 | 0.41±0.02 | **0.04**±0.00 | 3.74±0.09 |
| HTFlows | **0.90**±0.00 | 0.39±0.02 | **5.63**±0.15 | 0.42±0.06 | 0.49±0.04 | 0.07±0.00 | **2.97**±0.31 |

Table A5: Full benchmark metrics for random generation using ZINC-250K (reporting mean±std).

|  | Validity ↑ | Uniqueness ↑ | Novelty ↑ | SNN ↑ | Frag ↑ | Scaf ↑ | IntDiv$_1$ ↑ |
|---|---|---|---|---|---|---|---|
| Data (ZINC-250K) | 1.00±0.00 | 1.00±0.00 | 0.02±0.00 | 0.51±0.00 | 1.00±0.00 | 0.28±0.02 | 0.87±0.00 |
| GraphDF | - | **1.00**±0.00 | **1.00**±0.00 | 0.23±0.00 | 0.35±0.01 | 0.00±0.00 | 0.88±0.00 |
| MoFlow | **0.70**±0.01 | **1.00**±0.00 | **1.00**±0.00 | **0.29**±0.00 | **0.81**±0.01 | **0.01**±0.00 | 0.88±0.00 |
| HTFlows | 0.46±0.02 | **1.00**±0.00 | **1.00**±0.00 | 0.22±0.00 | 0.57±0.03 | 0.00±0.00 | **0.90**±0.00 |

|  | IntDiv$_2$ ↑ | Filters ↑ | FCD ↓ | logP ↓ | SA ↓ | QED ↓ | Weight ↓ |
|---|---|---|---|---|---|---|---|
| Data (ZINC-250K) | 0.86±0.00 | 0.59±0.01 | 1.44±0.01 | 0.05±0.01 | 0.03±0.01 | 0.01±0.00 | 2.18±0.39 |
| GraphDF | 0.87±0.00 | **0.54**±0.01 | 34.30±0.30 | 1.28±0.03 | 1.70±0.03 | 0.30±0.00 | 149.27±1.55 |
| MoFlow | 0.86±0.00 | 0.53±0.02 | **22.65**±0.40 | **0.14**±0.03 | **0.85**±0.04 | 0.24±0.01 | 61.83±3.00 |
| HTFlows | **0.88**±0.00 | 0.22±0.02 | 27.90±0.23 | 0.96±0.07 | 2.07±0.05 | 0.44±0.01 | **16.51**±2.85 |

Then we find molecule candidates $\{G_i\}_{i=1}^k$ with top-$k$ properties in the data set $\mathcal{D}$ are chosen. New embeddings are explored by optimizing the predict label by $g_{\theta*}$ starting from these candidates:

$$\boldsymbol{h}_{i,j} = \delta\frac{\partial g_{\theta*}}{\partial \boldsymbol{h}}(\boldsymbol{h}_{i,j-1}) + \boldsymbol{h}_{i,j-1}, \quad j = 1,\dots,N, \quad \boldsymbol{h}_{i,0} = f(G_i), \quad i = 1,\dots,k,$$

where $\delta$ denotes the search step length, and $N$ is the number of iterations. These embeddings could be recovered to be molecule set:

$$\mathcal{D}' = \{f^{-1}(\boldsymbol{h}_{ij})\}_{i=1,\dots,k, \quad j=1,\dots,N}.$$

Finally, $\mathcal{D}'\backslash\mathcal{D}$ gives the novel molecule sets with related high target properties.

**Generation results** In our experiments, the $g_\theta$ is a simple 3-layer MLP with 16 hidden nodes, the dataset $\mathcal{D}$ is ZINC-250K, and target property $y$ is QED. And $\mathcal{D}'\backslash\mathcal{D}$ provides 17 molecules with QED score 0.948. The Top-3 QED score and molecular SMILES are listed below:

1. QED = 0.948442, `CC(C)N1N=CC2=NC(c3ccc(-c4ccccn4)cc3)NC21`

2. QED = 0.948190, `O=C(NCC1COc2ccccc2O1)c1ccccc1Cl`

3. QED = 0.948051, `Cc1ccc(C(CO)C2CS(=O)(=O)c3ccccc32)cc1`

**Baselines** The baselines scores of GCPN (You et al., 2018), GraphAF (Shi et al., 2019), GraphDF (Luo et al., 2021), MoFlow Zang & Wang (2020) and ModFlow (Verma et al., 2022) are acquired from the corresponding papers. The score of JTVAE (Jin et al., 2018) is acquired from Zang & Wang (2020); Verma et al. (2022).

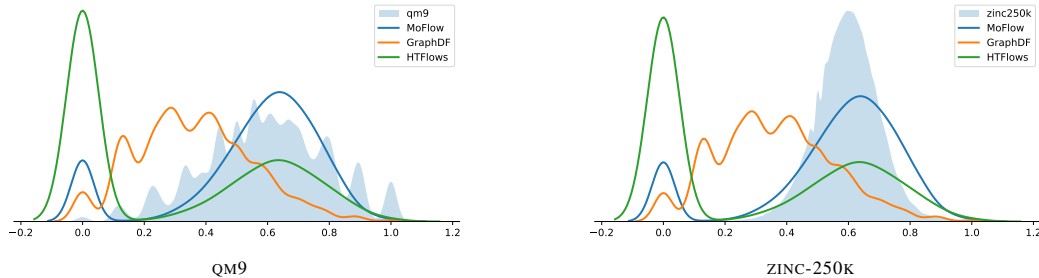

Figure A9: Node homophily distribution of generated molecules.

