# OpenReview forum: "Molecule Generation by Heterophilious Triple Flows"
_ICLR.cc/2024/Conference — Submitted to ICLR 2024_

### Official Review · Reviewer_VenB · 2023-10-28

**Soundness:** 2 fair
**Presentation:** 3 good
**Contribution:** 3 good
**Rating:** 6
**Confidence:** 3

**Summary:**

This paper introduces a new framework called Heterophilous Triple Flows (HTFlows) for generating molecules with desired properties. It discusses the limitations of existing approaches, such as graph neural networks with heterophily, and proposes HTFlows as a solution to address these issues and improve performance on chemoinformatics benchmarks.

**Strengths:**

The presentation is clear.

The contribution is good if more evidence is provided

**Weaknesses:**

See below

**Questions:**

1. Researchers already find that heterophily is not always harmful and homophily assumption is not always necessary for GNNs [1,2,3,4]. How does this paper align with these works?

2. Heterophily was usually studied in node classification tasks when graph-aware models underperform graph-agnostic models. I'm not sure if heterophily will also cause performance degradation of graph-aware models in generative tasks. Do you have any evidence or references? If the answer is yes, it would be a good contribution.

3. In equation (2), what is S?

4. Ablation study is missing. How does each component in heterophilious message passing contribute to the performance gain?

If the authors can answer the above questions well, I will raise my score.


[1] Is Homophily a Necessity for Graph Neural Networks?. In International Conference on Learning Representations 2022.

[2] Revisiting heterophily for graph neural networks. Advances in neural information processing systems, 35, 1362-1375.

[3] When do graph neural networks help with node classification: Investigating the homophily principle on node distinguishability. arXiv preprint arXiv:2304.14274.

[4] Demystifying Structural Disparity in Graph Neural Networks: Can One Size Fit All?. arXiv preprint arXiv:2306.01323.

---

> ### Author Response · Authors · 2023-11-20
> **Reply to review VenB**
>
> Thank you for providing constructive feedback that has allowed us to clarify essential aspects of this paper. We believe our response has adequately responded to your concerns.
>
> **Q1:** *There are many literatures showing that heterophily is not always harmful. How does my paper align with these?*
> **A1:** We thank the reviewer for pointing out these works. We are amending the related work section to include these works. We summarize the relation to these works as follows:
>
> * Paper [1] points when the cross-class neighbour similarity is stronger than intra-class similarity, which keeps the post-aggregated embeddings of different classes distinguishable, then the data heterophily is not that harmful for node classification.
> * Paper [2] claims their newly designed aggregated homophily is a good indicator of the model expressiveness. However, it’s not very surprising that this homophily indicator is highly correlated with the performance of the model on certain data, since it combines the information from the data (node features, aggregation operator and labels) and model structure (assumption on model type: simplifying graph convolutional networks). In the model design, we share a similar idea: trying to process lower/higher frequency information separately in different channels. At the same time, we are different: For example, while handling high-frequency info, we avoid the information exchange between highly different nodes but keep smoothing the similar clusters. ACM capture high-frequency data by the graph Laplacian, which cares more about the graph topology (this idea comes from graph signal processing)
> * Paper [3] verifies that compared with homophily, GNN performance on node classification is more related to intra- and inter-class Node Distinguishability (ND). In molecular generation, graph distinguishability is much more important than ND.
> * Paper [4] defines the majority pattern (homophilic nodes in homophilic graph/ heterophilic nodes in the heterophilic graph) and shows that GNN is superior to graph-agnostic models on the majority patterns but worse on minority patterns. It tries to reveal how does the heterophily influence the GNN performance on node classification.
>
> In general, all these papers [1,2,3,4]:
> * Focus on the deeper insights between homophily and the GNN model performance on the node classification task. The heterophily matters in different styles: the difference with node classification and graph generation is discussed at the general reply.
> * They try to discover the more essential factor (compared with heterophily) which affects the model performance. At the same time, we develop our methods, motivated by noticing the possible lossing information during training from the inspiration of heterophily.
>
>
>
> **Q2:**  What is the problem of data heterophily in molecular generation tasks? any evidence or reference?
> **A2:** Key to the generative model is learning the data distribution, which possibly contains special patterns underlying the data domain. Here the data is graph-structured, so an expressive model can distinguish different graphs. The smoothing embeddings from the message passing scheme will decrease the expressiveness of models. Thus it is natural to build structure to mitigate this problem. This is the motivation for us to design different flows to capture more information and avoid missing certain types of information. We have included an example analysis in the general reply to all reviewers.
>
> **Q3:** In equation (2), what is $S$?
> **A3:** In the ACLs (affine coupling layers), the input $X$ is split to two parts $X_1, X_2$ by the masking matrix. One part $X_1$ is updated by another part $X_2$ in the following style. For reversibility, the transformation is chosen to be an affine map, then $X_1$ is updated by the factorization matrix $S$ and translation matrix $T$ to be $SX_1+T$. And $S, T$ are generated from $f(X_1)$, where the $f$ is the coupling function. To avoid the negative factor, we choose $\log S, T = f(X_1)$, as mentioned in Equation (2). The coupling function $f$ is defined to be the GNN with heterophilous message passing layer in Equation (9): $ACL^{(GNN_i^\gamma, \mathbf{M}_i)}$
>
> **Q4:** *Ablation study is missing. How does each component in heterophilious message passing contribute to the performance gain?*
> **A4:** Our model adds structure caring about heterophily/homophily based on MoFlow. And MoFlow utilizes the traditional GNN so loses the high-frequency information on heterophilic data. Thus MoFlow could be viewed as a single flow (HTFlows without heterophily and homophilic flows) as ablation. And there is also an example analysis in the general reply.
>
> **References**  the same as the reviewer given.

---

> > ### Comment · Reviewer_VenB · 2023-11-22
> >
> > Thanks for the response. The authors have addressed most of my concerns and I think studying heterophily problem for generative models is a very interesting topic. Thus, I will increase my rating to 6. Good luck.

---

> > > ### Author Response · Authors · 2023-11-23
> > > **Grateful for your support**
> > >
> > > We are glad to note that our response adequately addressed your concerns, and share your enthusiasm about the merits of investigating heterophily in the context of generative models.  Thank you so much!

---

### Official Review · Reviewer_LyRR · 2023-10-31

**Soundness:** 1 poor
**Presentation:** 2 fair
**Contribution:** 1 poor
**Rating:** 3
**Confidence:** 4

**Summary:**

This paper proposes the heterophyllous triple flow model to handle the heterogeneity of molecular graph generation. Its key idea is to introduce multiple interactive flows which "capture" heterophily patterns in the molecular space.

**Strengths:**

This paper tackles the problem of generating heterophilious molecular graphs, where vertices may have different features (e.g., atom types) even when they are adjacent to each other.

**Weaknesses:**

### Weak experiments

My main criticism is that the experiments are not enough to verify the practical relevance of the proposed work.

The authors seem to consider the baselines proposed by Verma et al., 2022 as state-of-the-art. However, there exists a plethora of molecular generative models since the work of Verma et al., 2022. Just to list a few examples, one could consider STGG (Ahn et al., 2022), GDSS (Jo et al., 2022), Digress (Vignac et al., 2022), and GraphARM (Kong et al., 2023). The authors could even consider SMILES-LSTM (which demonstrates surprisingly good performance) for more comprehensive baselines.

### Lack of justification

I was unable to find a good justification for why the proposed flow network better generates heterophilious graphs. The only explanation I got was that "binary masking ensures that only part of the input is transformed, allowing the model to retain certain features while altering others, enabling the flow to capture intricate data distribution characteristics". I do not understand why retaining certain features is related to "capturing intricate data distribution characteristics".

**Questions:**

I think one could easily incorporate the heterophilious nature by parameterizing molecular generative models with GNNs specifically designed to mitigate over smoothing and better recognize heterophilious graphs. Could the authors provide explanation on why simply using such heterophilic GNNs cannot resolve the considered issue?

---

> ### Author Response · Authors · 2023-11-20
> **Reply to review LyRR**
>
> We appreciate your valuable feedback, which has assisted us in clarifying crucial elements of this paper. We trust that our reply could adequately deal with your apprehensions and results in an improved rating
>
> **Q1:** *Weak experiments: more SOTA experiments should be included.*
> **A1:** Many of the SOTA methods do not have code available for reproducing the runs to generate the chemoinformatics metrics. Also copying over partial results from the papers directly does not seem feasible or fair as the evaluation setups differ. For example, in STGG (in the appendix) FCD-qm9 is 0.58 and FCD-zinc is 0.2778, which are even better than the ground truth values we get. They also utilised chemical rules in the learning process, while we hope to learn the data distribution without hand-tailoring this prior knowledge.
>
> Like most of the methods, GDSS, DiGress and GraphARM don’t provide the comprehensive cheminformatics metrics evaluation on QM9 or ZINC, which brings difficulties to merge the results. The main baseline for our work is MoFlow, on which we demonstrate the benefits of the heterophilous flow approach and show to outperform it.
>
> **Q2:** Why the proposed flow network better at generating heterophilious graphs?
> **A2:** Our model is not aiming to be better at generating heterophilious graphs, but to avoid lose information from heterophilious data points during training. Molecular generation cares more about distribution learning. Please see our general reply to all reviewers, where there is an example analysis about problems on heterophilous data and how our model benefits.
>
> **Q3** Why not just utilize heterophilic GNNs but design this structure?
>
> **A3** The difference between node classification and graph generation task are discussed at general reply. For molecular generation task, graph distribution learning requires avoiding any possible information loss. The existing heterophilic GNN could mitigate oversmoothing problem for node classification. The heterophilic GNNs are good at making heterophilic nodes distinguishable. But it’s not enough for simply replacing traditional GNN by it in current problem scenorias, because it may ignores the subtle low-frequency in the homophilic data. Keeping both traditional GNN (in central flow) and extra two flows helps the model get distinguishable embeddings (from both low and high frequency) for all types of graph-structured data points.

---

> > ### Comment · Reviewer_LyRR · 2023-11-23
> >
> > Thank you for the detailed rebuttal.
> >
> > Q1. I resonate with the authors on the difficulties of reproducing the baselines. However, the baselines, e.g., Digress, have open-source implementation. The authors could have run the code to obtain the molecules and measure the cheminformatics metrics. At least, the authors could have compared their method with SMILES-LSTM.
> >
> > Q2. I believe the interactions described by the authors can already be captured by using a regular GNN. It is an oversimplification to describe GNN as a simple heat conduction process, and I think the authors should provide a more formal description of "avoiding losing information from heterophilious data points" to be persuasive.
> >
> > Q3. Unfortunately, I could not fully understand your argument. However, from your argument, it seems that combining traditional GNNs with the ones developed for mitigation of oversmoothing would solve the information loss problem. However, my main concerns are more on Q1 and Q2.

---

> ### Author Response · Authors · 2023-11-23
> **Thank you - please see our response**
>
> Thank you for your comments.
>
> 1) Please note that the baseline MoFlow we compared with is known to strongly outperform baselines such as RNNs/LSTMs and advanced flow models such as GraphAF, and methods such as Junction tree VAEs etc. Since we improve upon MoFlow, we inherit the benefits with respect to these baselines.
>
> 2, 3) Please see our global response, where we give a concrete example of how GNNs fail to take into account the effect of heterophily in generative settings, and how our modelling helps overcome this issue.
>
> Thank you again for your feedback, and we hope this addresses your concerns.

---

> > ### Comment · Reviewer_LyRR · 2023-11-23
> >
> > Thank you for the swift reply and sorry for being late to engage with the reviewer discussion phase.
> >
> > I am still not very convinced by the provided arguments, but I will try to understand them, further discuss them with other reviewers, and adjust my scores accordingly.

---

### Official Review · Reviewer_AKoh · 2023-11-01

**Soundness:** 2 fair
**Presentation:** 3 good
**Contribution:** 2 fair
**Rating:** 5
**Confidence:** 3

**Summary:**

This paper aims to tackle the problems of using GNNs for molecule generalization under the heterophilious input setting, where some conventional GNNs could fail on this setting due to their strong homophilious assumption. Specifically, the paper designs a new GNN model with three interactive flows to capture heterphiliy patterns in the molecular space. The effectiveness of the proposed model are validated by the experiments on several benchmark datasets for molecule generation and modelling.

**Strengths:**

1. The paper is clear and well-structured.

 2. The proposed ACL blocks in the model are shown to be inherently inversable.

**Weaknesses:**

1. The level of homophily in a graph is defined based on node labels in GNN literature, where high homophily is observed when neighboring nodes share the same labels, and vice versa. However, In the design of the paper’s heterophilious message passing layer, i.e., equation (6), they define the homophily of nodes as the cosine similarity between pair node embeddings. This could be problematic since the cosine similarity between pair node embeddings might not align with their labels. As a result, the proposed model may inherently fail to work well in the cases that the cosine similarity between pair node embeddings is not aligned with their labels.

 2. Numerous GNN architectures have been developed for heterophilic graphs, where they have been demonstrated their effectiveness for heterophilious graphs comes from their ability to work as high-pass filters. However, there is no solid justification indicating that the proposed heterophilic flows can effectively handle heterophilic graphs. Additionally, it is also not clear that the benefits of the designed heterophilic flows, as compared to directly adapting existing heterophilic GNN structures for molecular generalization. More discussion here would be helpful.

 3. Lack of ablation studies on different components. The proposed model consists of several components, including bond flow and heterophilious atom flow. Moreover, the heterophilic atom flow encompasses three interacting flows: the central, homophilic, and heterophilic flows. It remains unclear which component is most crucial or how each contributes to the model's overall performance. A detailed breakdown and analysis would provide greater clarity.

 4. The paper claims that existing GNN models for molecule generalization have overlooked the repulsions between dissimilar atoms and are vulnerable to oversmoothing. However, it is not evident that the proposed model effectively addresses the oversmoothing issue. It would be better to provide more explanations on this and conduct experiments to validate the oversmoothing claim.

 5. Given that this study seeks to address the challenges of heterophily and oversmoothing in GNNs for molecular generalization, it would be beneficial to delve deeper into papers on GNNs concerning heterophilious graphs and oversmoothing problems in the related work section.

**Questions:**

Please refer to the weaknesses.

---

> ### Author Response · Authors · 2023-11-20
> **Reply to review AKoh**
>
> Thank you for your constructive feedback that helped us elucidate some important aspects of this work. We hope our response has sufficiently addressed your concerns, and translates into an increased score.
>
> **Q1:** *The potential insonsistency between node embeddings and labels. The reviewer finds defining cosine similarity as homophily is problematic.*
> **A1:** This inconsistency appears in the node classification task as discussed in the general reply. Here the node features are the one-hot encoding of the node label (atom type). Since the node label is the learning target during the molecule generation, we need to design a substitute way to approximate the homophily/heterophily. The flow maps the sampled Gaussian embeddings back to the one-hot encoding of atom type (label), to be consistent with the definition of homophily of the final layer, we extend this concept to the intermediate layers as the cosine similarity between the node embeddings.
>
> 1. The node features $x_v$ of node $v$ is initialized by the one-hot encoding of atom type. we can write this to be  this node
> $$ x_v = [1_{v=C}, 1_{v=N}, 1_{v=O}, 1_{v=F} ] $$
> e.g. for atom C, the node feature is $[1,0,0,0]$.
>
> 2. In the flow, the node features are mapped into latent space with a Gaussian distribution. During the molecular generation process, we sample Gaussian distributed random vectors and map them back to node features at the final layer, then this feature is transferred to a node label. It means we don't know the node label until the final layer. To define the homophily for a node in the intermediate layer without a label, we choose the similarity between the half-transferred embeddings to be the homophily. This definition style is also consistent with the final layer.
>
> **Q2.1:** *Need to show the designed heterophilic flows expressive on heterophilic graphs.*
> **A2.1:** This is not a node classification task but a generation task. It could be easy to show the ratio of the heterophilic nodes is correctly classified. But here for the generation task, the model is supposed to learn the data distribution, with as least missing information as possible. Traditional GNNs will smooth the high-frequency information on the graph easily. We provide example analysis and discussion in the general answer.
>
> **Q2.2:** *Why design the heterophilic flows but not directly adapt existing heterophilic GNN?*
> **A2.2:** The difference between node classification and graph generation task are discussed at general reply. For molecular generation task, graph distribution learning requires avoiding any possible information loss. The existing heterophilic GNN could mitigate oversmoothing problem for node classification. The heterophilic GNNs are good at making heterophilic nodes distinguishable. But it's not enough for simply replacing traditional GNN by it in current problem scenorias, because it may ignores the subtle low-frequency in the homophilic data. Keeping both traditional GNN (in central flow) and extra two flows helps the model get distinguishable embeddings (from both low and high frequency) for all types of graph-structured data points.
>
>
>
> **Q3:** *The ablation study on different flows suggested.*
> **A3:** Our model adds structure caring about heterophily/homophily based on MoFlow. And MoFlow utilizes the traditional GNN so loses the high-frequency information on heterophilic data. Thus MoFlow could be viewed as a single flow (HTFLows without heterophily and homophilic flows) for ablation study.
>
> **Q4:** *How does the model overcomes the issue of oversmoothing?*
> **A4:** There is disscusion and example analysis in the general reply.
>
> **Q5:** *Suggestion: more content about heterophily in the related work section.*
> **A5:** Following reviewer **VenB**'s comments, we are amending the related work section with more details and background on heterophily.

---

### Official Review · Reviewer_x2it · 2023-11-06

**Soundness:** 3 good
**Presentation:** 3 good
**Contribution:** 2 fair
**Rating:** 5
**Confidence:** 3

**Summary:**

This paper proposes HTFlows, a flow-based method for molecular graph generation. It addresses heterophily in molecules while existing molecular graph generation methods using graph neural networks make a homophily assumption that neighboring nodes have similar features. HTFlows uses multiple interactive normalizing flows to model homophilic, heterophilic, and central node patterns to capture nuanced molecular dependencies. Extensive experiments benchmark performance on QM9 and ZINC-250K datasets in molecule generation and property optimization tasks. Key results show HTFlows achieves high validity without extra checks, optimizes target properties well, and generates high-quality diverse molecules.

**Strengths:**

This paper effectively addresses the challenge of modeling heterophily in molecular graphs, a problem that challenges conventional homophily-based approaches. The proposed interactive multi-flow architecture enables the capture of nuanced molecular patterns across varying homophily-heterophily levels, enhancing versatility in representation. The paper rigorously evaluates the proposed method across various metrics on standard molecule datasets. The performance of HTFlows is comprehensively demonstrated by comparing it to state-of-the-art baselines like GraphDF and MoFlow.

**Weaknesses:**

The proposed HTFlows only brings improvements on limited metrics when compared to state-of-the-art baselines (as listed in Table A4, A5, and Table 3), which constrains its contribution and impact. Besides, it is unclear about the connections between the improved metrics and the introduced heterophilious triple flows. It would be more convincing to bring theoretical analysis and ablation study to demonstrate the effect of emphasizing heterophily.

**Questions:**

1. Why the mixing of ACL in the heterophilous atom flow are performed one by one for each atom type rather than in parallel?
2. It seems that the histograms in Fig. 6 and 7 do not match the results in Table A4 and A5. For example, why the molecular weights in Fig.6 are mainly 75~175, but the mean molecular weights in Table A4 are much smaller? Besides, as shown in Figure 7, the molecular weights of GraphDF seem to have a significantly lower mean value than MoFlow and HTFlows based on their histograms, but HTFlows has the lowest mean molecular weight as listed in Table A5.

---

> ### Author Response · Authors · 2023-11-20
> **Reply to reviewer x2it**
>
> We thank the reviewer for their comments. We reply to each concern/question below in the order received.
>
> **Q1:** *Limited improvement on all metrics.*
> **A1:** We focus on showing the benefits of our HTFlows framework over MoFlow, which can be seen as a base. Rather than cherry-picking just a few metrics where we excel, we on purpose show a wide range of chemoinformatics metrics. Some of the metrics are a bit of a trade-off; improvement on one metric can have a negative influence on other metrics. For example, if a model would always generate molecules with only carbons, it will get 100% validity, but the diversity should be low. Especially on QM9 (that other works typically test on), we achieve very good performance overall.
>
> **Q2:** *What are the connections between the heterophily and the improved metrics?*.
> **A2:**  Generative model aims to learn the distributions related to several important patterns from the dataset. Most GNN-based models fail to capture enough information from the heterophilic graph data since the message-passing scheme smooths the graph embeddings after several layers. These improved metrics are good indicators showing the better distribution learning of our model.
>
> **Q3:** *The HTFlows' structure: why design mixing ACL iteratively but not in parallel?*
> **A3:** The different flows learn different types of information. We design the interactions for different channel so that they could also utilize the information from other patterns. However, for the reversibility, ACL updates one part of data by another part, which means there is always an order for which part is updated (impossible to update all at the same time, otherwise the structure is not reversible anymore). More layers of the current design can also make sure enough interactions compared to parallel design.
>
> **Q4:** *Inconsistency between weight histogram (Fig 6,7) and tables (Table A4, A5)*
> **A4:** They are consistent. As described in App. C, the metrics in Table A4–A5 are the Wasserstein distance/discrepancy metric between the generated molecules and the original data set, summarizing the histogram result into one number.

---

> > ### Comment · Reviewer_x2it · 2023-11-23
> >
> > Thanks for the authors' response that addresses my concerns and questions. I have increased my score to 5 based on the revised version.

---

> > > ### Author Response · Authors · 2023-11-23
> > > **Thank you - glad that your concerns and questions have been addressed.**
> > >
> > > Many thanks for your feedback and confirming that our response has addressed your concerns and questions. We are grateful for your improved assessment, and would greatly appreciate your stronger support for the paper during the AC-reviewer discussion phase so that the merits of incorporating heterophily in generative settings and its promise for important applications such as drug discovery can be communicated to our wider ML community. Thank you very much!

---

### Author Response · Authors · 2023-11-20
**General answer to all reviewers**

We thank all the four reviewers for their insights and comments. The reviewers found the paper rigorous, clear, and well-structured. We have replied to each individual review in separate messages, but stress the points raised by reviewers here.

**1. Discussion and example about how our model avoids information loss from heterophilic data during training**

Heterophily is an important factor which influences GNNs performance in general, especially in molecule modelling. Our heterophilous structure can collect more meaningful information from graphs, compared with the traditional GNN-based models. The following gives a practical/simple example of how the model avoids heterophily-related problems:

* Consider the molecule "Phenol" (`C6H5OH`) that consists of a phenyl group (`−C6H5`) bonded to a hydroxy group (−OH). In the graph view, the nodes set (`1.C 2.C 3.C 4.C 5.C 6.C 7.O`), edge set (`1-2, 2-3, 3-4, 4-5, 5-6, 6-1, 6-7`). Assume the node feature for Carbon is 0, and the node feature for Oxygen is 1, then the node features are (`0 0 0 0 0 0 1`).
* In $GNN^{cen.}$ (GNN for the central flow, traditional GNN), the whole message-passing process is similar to a heat conduction process. After enough steps, the information will spread out evenly over the graph.
* In $GNN^{hom.}$ (GNN for the homogeneous flow, which allows more information exchange from nodes with similar features), the message-passing process avoids information exchange between the two clusters (the phenyl group with six Carbons and single hydroxy group), since the only edge (6-7) between two clusters is blocked because the two end-nodes of it are orthogonal.
* In $GNN^{het.}$ (GNN for the heterogeneous flow, which allows less information exchange from nodes with similar features), in the first layer of message passing, only the edge (6–7) bridging the two groups transfers the information from 7Oxygen to 6Carbon. In the second layer, node 6Carbon gets some energy from the last layer, which differs from other carbon neighbours (1,5), then spreads the message to them, at the same time, since 6Carbon and 7Oxygen become more similar after the last layer. Thus, they transfer less information on the 2nd layer compared to the 2nd layer of $GNN^{cen.}$. This is more like heat conduction with speed control, which decreases the conduction speed on smooth areas.
* With three flows having different message passing styles, our model becomes more expressive compared to the structure that only considers the traditional message passing in $GNN^{cen.}$, which is very similar to the heat conduction process. For heterophilous data, the $GNN^{het.}$ keeps the high frequency appearing in the topology as much as possible, at the same time, $GNN^{hom.}$ keeps the low frequency from not disappearing, through the resistance on information exchange between similar features.

Our work learns to model these kind of phenomena by capturing more general heterophily through the node embeddings. The approach is expressive and novel, and we show that it has practical benefits.

**2. Difference between node classification and graph generation**

Many prevoius work on heterophily discussed the model performance on node classification tasks. In most problem settings of node classification, the embeddings of nodes (output of GNN model) are used for the classification, and the node features are assumed to be correlated with the real labels (often sampled from distribution dependent on node label).

The graph generation task requires models to learn the data distribution, which prefers high graph distinguishability. Here the node labels are decided by the features, which are more stable compared with the node classification task settings. With the analysis in the general reply, the heterophilic graph data become less distinguishable after message passing. To design more expressive models, the model is required to separate different graph points (distinguish all types of graphs, including homophilic and heterophilic ones), thus we need to care about this issue.

---

> ### Author Response · Authors · 2023-11-22
> **Submission updated (related work)**
>
> We have now updated our submission to include the promised changes in the literature review (see Section 2 Related Work).

---

### Meta-Review · Area_Chair_6Mmk · 2023-12-06

**Metareview:**

The paper tackles the implicit bias of GNN-based generative models to generate graphs that have more homophilic (similar) nodes. Specifically, the Authors propose a flow-based generative model that is designed to avoid such tendency.  One shared concern among some of the reviewers was the lack of clarity on why the proposed method allows generating molecules with more diverse (heterophylic) atoms. The reviewers pointed out lack of specific ablations and unclear motivation behind the design of the model. These concerns were not fully alleviated during the rebuttal phase. Another important issue is the lack of comparison to a broader set of generative methods. The Authors have focused largely on flow-based methods, which represents only one family of models used for similar tasks.  Based on these and other concerns raised by reviewers, I have to unfortunately recommend rejection at this stage.

**Justification For Why Not Higher Score:**

The paper doesn't give sufficient clarity on the main novelty of the paper, including the improvement over relevant baselines.

**Justification For Why Not Lower Score:**

N/A

---

### Decision · Program_Chairs · 2024-01-16

Reject